# Fatty Acid-Binding Protein 4—An “Inauspicious” Adipokine—In Serum and Urine of Post-Partum Women with Excessive Gestational Weight Gain and Gestational Diabetes Mellitus

**DOI:** 10.3390/jcm7120505

**Published:** 2018-12-02

**Authors:** Żaneta Kimber-Trojnar, Jolanta Patro-Małysza, Marcin Trojnar, Katarzyna E. Skórzyńska-Dziduszko, Jacek Bartosiewicz, Jan Oleszczuk, Bożena Leszczyńska-Gorzelak

**Affiliations:** 1Chair and Department of Obstetrics and Perinatology, Medical University of Lublin, 20-090 Lublin, Poland; jolapatro@wp.pl (J.P.-M.); jacek.bartosiewicz@umlub.pl (J.B.); jan.oleszczuk@umlub.pl (J.O.); b.leszczynska@umlub.pl (B.L.-G.); 2Chair and Department of Internal Medicine, Medical University of Lublin, 20-081 Lublin, Poland; marcin.trojnar@umlub.pl; 3Chair and Department of Human Physiology, Medical University of Lublin, 20-080 Lublin, Poland; katarzyna.skorzynska-dziduszko@umlub.pl

**Keywords:** adipokines, fatty acid binding protein 4, leptin, gestational diabetes mellitus, excessive gestational weight gain, bioelectrical impedance analysis, body composition, hydration status, type 2 diabetes mellitus

## Abstract

The exact roles of adipokines in the pathogenesis of type 2 diabetes and obesity are still unclear. The aim of the study was to evaluate fatty acid binding protein 4 (FABP4) concentrations in the serum and urine of women with excessive gestational weight gain (EGWG) and gestational diabetes mellitus (GDM) in the early post-partum period, with reference to their laboratory test results, body composition, and hydration status. The study subjects were divided into three groups: 24 healthy controls, 24 mothers with EGWG, and 22 GDM patients. Maternal body composition and hydration status were evaluated by the bioelectrical impedance analysis (BIA) method. Concentrations of FABP4, leptin, and ghrelin were determined via enzyme-linked immunosorbent assay (ELISA). Healthy women were characterized by the lowest serum leptin concentrations and by a negative correlation between the serum and urine FABP4 levels. Serum FABP4 levels were the highest in the GDM group. Serum FABP4 and leptin concentrations correlated positively in the GDM group. The EGWG group had the highest degree of BIA disturbances in the early puerperium and positive correlations between the urine FABP4 and serum leptin and ghrelin concentrations. The physiological and pathological significance of these findings requires further elucidation.

## 1. Introduction

According to the current state of scientific knowledge, pre-pregnancy obesity and excessive gestational weight gain (EGWG) as well as gestational diabetes mellitus (GDM) have been regarded as independent risk factors for post-partum complications in both mothers and their children, which will subsequently lead to an increased risk of chronic diseases later in their lives [1,2,3,4,5]. It has been observed that women with a previous history of GDM and EGWG are much more prone to suffer from type 2 diabetes mellitus (T2DM), obesity, cardiovascular diseases (CVDs), and metabolic syndrome in the future [1,2,3,4,5].

The lifestyle of pregnant women depends on the rate and extent of their gestational weight gain, which may result in the persistence of overweight and obesity in the postnatal period and the occurrence of new complications, including T2DM and metabolic syndrome. Gestational weight guidelines of the Institute of Medicine [6] provide ranges of recommended weight gain for specific pre-pregnancy body mass index (BMI) categories in relation to the least risk of adverse perinatal outcomes. It is recommended that in order to prevent adverse maternal, as well as infant, outcomes, women with normal weight at the time of conception should limit their total weight gain in pregnancy to 11.5–16 kg, overweight women to 7–11.5 kg and obese women should not exceed the weight gain of 5–9 kg [6].

Compared to women with healthy pregnancies, mothers with histories of GDM have elevated metabolic syndrome risk factors, including higher blood pressure, dyslipidemia, and obesity. Up to one-third of women with T2DM have previously been diagnosed with GDM [7,8]. Women who had prior GDM have a 36–70% risk of developing T2DM later in life, depending on risk factors and length of follow-up [9]. Nearly half of GDM patients will develop to T2DM and obesity in the future [10].

In this study, we concentrated on the fatty acid binding protein 4 (FABP4). This newer adipokine appears to be one of the most probable candidates involved in the pathophysiology of GDM, similarly to well-known and verified markers of diabetes and obesity, such as adiponectin, leptin, and tumour necrosis factor α (TNFα) [11,12,13]. Furthermore, the roles of FABP4 in the development of adiposity, insulin resistance, T2DM, atherosclerosis, hypertension, coronary artery and cerebrovascular diseases, as well as metabolic syndrome, are of great concern [14,15,16,17,18].

FABP4, also known as adipocyte FABP (A-FABP), is a member of a lipid-binding protein super-family. FABP4 is highly expressed in adipocytes and consists of about 1% of all soluble proteins in the adipose tissue [19]. Expression of FABP4 is highly induced during adipocyte differentiation and transcriptionally controlled by peroxisome proliferator-activated receptor γ (PPARγ) agonists, unsaturated long-chain fatty acids, dexamethasone, and insulin-like growth factor 1 [19,20,21]. FABP4 is also detected in a subset of the endothelial cells and macrophages, where it increases accumulation of cholesterol ester and foam cell formation via inhibition of the PPARγ-liver X receptor α (LXRα)-ATP-binding cassette A1 (ABCA1) pathway and induces inflammatory responses through activation of the IKK-NF-κB and JNK-AP-1 pathways [21]. The intracellular FABP4 contributes to the fatty acid uptake and intracellular transfer. Secretion of FABP4 from adipocytes is induced by lipolytic agonists or nutrient deprivation, and this secreted form is proposed to control glucose production by hepatocytes and insulin secretion by the pancreatic β-cells [20,22].

FABP4 is partially secreted by microvesicles derived from adipocytes (an established mechanism for unconventional secretion from adipocytes), and both microvesicle-free-mediated and microvesicle-secreted FABP4 are downregulated by insulin [21]. The fact that evidence supporting extracellular roles of FABP4 as an adipokine is piling up is of great importance. The secreted form of FABP4 may be biologically active, which could lead to a paradigm shift in the understanding of FABPs as lipid chaperones and the local networking of metabolic and inflammatory responses. Direct effects of exogenous FABP4 have been demonstrated in multiple types of cells; FABP4 has been reported to enhance hepatic glucose production in vivo and in vitro, decrease cardiomyocyte contraction in vitro, inhibit expression/activation of the endothelial nitric oxide synthase (eNOS) in vascular endothelial cells, and to increase proliferation/migration of the vascular smooth muscle cells and glucose-stimulated insulin secretion in the pancreatic β cells. It has been shown that FABP4 is secreted from adipocytes and may act as an adipokine [21].

However, the evaluation of FABP4 in the early post-partum period has not been performed so far. Furthermore, available studies on the urine FABP4 concentrations are limited to the patients with acute and chronic renal dysfunction [23,24,25]. As far as we know, the FABP4 levels in the urine of both GDM patients and EGWG women have not been investigated before.

The relationship between FABP4 and various biochemical and biophysical measurements in puerperal GDM and EGWG women is still awaiting explanation. We hypothesized that because of both hyperglycaemia first detected at 24–28 weeks of pregnancy and excess weight gain during pregnancy which are connecting with changes in the body composition and hydration status, the FABP4 concentrations in the serum and urine of women with GDM and EGWG in the early post-partum period would probably be impaired.

## 2. Experimental Section

The study comprised women who were in a singleton term pregnancy (after 37 weeks of gestation) and were hospitalized at the Chair and Department of Obstetrics and Perinatology, at the Medical University of Lublin. The data collection was performed between March 2016 and February 2017. All of the study subjects included in this study were Caucasian. We selected three groups of patients. The first group consisted of 24 healthy controls, i.e., women without any metabolic disorders and with three normal results of the 2-h-75 g-oral glucose tolerance test (OGTT) at 24–28 weeks of gestation. This subgroup had no concomitant diseases, received only vitamin-iron supplementation, and presented normal values of pre-pregnancy BMI (i.e., between 18.5 and 24.99 kg/m^2^), normal gestational weight gain (i.e., 11.5–16 kg) [6], and proper gestational age.

The second group included 24 patients with EGWG, i.e., with normal pre-pregnancy BMI (i.e., between 18.5 and 24.99 kg/m^2^), three normal results of the OGTT at 24–28 weeks of gestation and gestational weight gain of at least 20 kilograms. This subgroup had no concomitant diseases and received only vitamin-iron supplementation.

The third studied group consisted of 22 patients with diagnosed GDM who were on a diet and receiving insulin treatment. 27% of the GDM subjects were treated with intensive insulin therapy, while 73% of them were controlled with only one basal insulin injection per day. Diagnostic criteria for GDM were based on the OGTT at 24–28 weeks of gestation: fasting glucose ≥5.1 mmol/L (92 mg/dL), or one hour plasma glucose result of ≥10.0 mmol/L (180 mg/dL), or a two-hour plasma glucose result of ≥8.5 mmol/L (153 mg/dL) [26,27]. This subgroup had no concomitant diseases and received only vitamin-iron supplementation and anti-diabetic treatment.

The exclusion criteria were as follows: multiple pregnancy, chronic infectious diseases, current urinary infections, abnormal laboratory results (e.g., the complete blood count, urine test, creatinine, glomerular filtration rate (GFR) findings), metabolic disorders (such as polycystic ovarian syndrome; except those listed in the inclusion criteria for the studied groups), mental illness, cancer, liver diseases, cardiovascular disorders, foetal malformation, premature membrane rupture, intrauterine growth retardation, the presence of metallic prostheses, and pacemakers or cardioverter-defibrillators.

Anthropometric measurements and sampling were performed after a 6-h fasting in the early post-partum period (i.e., 48 h after delivery). The maternal body composition and hydration status were evaluated by means of the BIA method and with the use of a body composition monitor (BCM) (Fresenius Medical Care). The serum levels of albumin, hemoglobin A1c (HgbA1c), and lipid profile were measured by a certified laboratory. After centrifugation, all of the collected maternal serum and urine samples were stored at −80 °C. The concentrations of FABP4 (R and D Systems, Inc., Minneapolis, MN, USA), leptin (R and D Systems, Inc., Minneapolis, MN, USA) and ghrelin (Wuhan EIAab Science Co., Wuhan, China) in these materials were determined using commercially available kits and in compliance with the manufacturer’s instructions via traditional enzyme-linked immunosorbent assay (ELISA). The survey was performed in duplicate for each patient.

All of the patients were informed about the study protocol, and a detailed written consent was obtained from each patient who agreed to participate in the study.

The study protocol received approval from the Bioethics Committee of the Medical University of Lublin (no. KE-0254/221/2015 (25 June 2015) and no. KE-0254/348/2016 (15 December 2016)).

All of the values were reported as the median (interquartile range 25–75%). The differences between the three studied groups were tested for significance using the Kruskal-Wallis analysis of variance. The post-hoc analysis of differences between two groups were tested for significance. The Spearman’s coefficient test was used for the correlation analyses. The Benjamini-Hochberg correction for false positive results was used. All of the analyses were performed using the Statistical Package for the Social Sciences software (version 19; SPSS Inc., Chicago, IL, USA). A *p*-value of <0.05 was considered statistically significant.

## 3. Results

The comparative characteristics of the study groups, presented in Table 1 and Table 2, revealed that healthy women had the lowest BMI at delivery, post-partum BMI, the serum leptin concentrations (Figure 1), total body water (TBW), extracellular water (ECW), and fat tissue index (FTI), as well as the highest concentrations of low-density lipoprotein cholesterol (LDL). The largest gestational weight gain, gestational BMI gain (ΔBMI-1), BMI gain in the period from pre-pregnancy to 48 h after delivery (ΔBMI), and intracellular water (ICW) were observed in the EGWG mothers. The most advanced age, as well as the highest pre-pregnancy BMI values and the serum FABP4 concentrations (Figure 1), were characteristic of the GDM group (Table 1 and Table 2). Compared to the healthy patients, the EGWG mothers presented increased values of the lean tissue index (LTI) and body cell mass index (BCMI), whereas the GDM women had lower levels of total cholesterol and high-density lipoprotein cholesterol (HDL). The BMI loss at 48 h after delivery (ΔBMI-2) was higher in the EGWG group than in the GDM group (Table 1 and Table 2).

No significant differences were observed between the groups with regard to other analysed parameters (Table 1).

The minimum detectable dose of human leptin is typically less than 7.8 pg/mL. Due to the urine leptin levels obtained in the majority of the EGWG and GDM patients, which were below the threshold of the sensitivity of the ELISA test, the “urine leptin” parameter is not present in these two groups in Table 1 and Table 3.

Figure 1 compares the serum levels of FABP4 and leptin between the control, GDM and EGWG groups in the early post-partum period (i.e., 48 h after delivery).

Positive correlations were found between the serum FABP4 concentrations and pre-pregnancy BMI, total cholesterol, HDL, and LDL levels in the control group. Negative correlations between the serum FABP4 concentration and ΔBMI-2, as well as HgbA1c and the urine FABP4 levels, were observed only in the healthy mothers (Table 3). Figure 2 is a scatter plot diagram of the negative correlation between the serum and urinary FABP4 concentrations in the control subjects. Positive correlations were found between the urine FABP4 concentrations and ΔBMI-2 and urine leptin level in the healthy patients. The urine FABP4 concentration correlated negatively with BMIs before pregnancy, at and after delivery, as well as with ΔBMI, albumin, and LDL levels, and all the studied BIA variables with the exception of ECW in the healthy group (Table 3).

Moreover, there were negative correlations between the serum FABP4 concentration and gestational weight gain, ΔBMI-1, BMI after delivery, ΔBMI, and ECW in the EGWG group. A direct correlation was found between the serum FABP4 and urine ghrelin concentrations. Positive correlations were present between the urine FABP4 and serum ghrelin and leptin levels in this group. The urine FABP4 concentration correlated negatively with the LTI and BCMI in the EGWG group (Table 3).

The serum FABP4 concentration correlated positively with the BMIs at and after delivery, serum leptin level, TBW, ICW and FTI in the GDM group. A negative correlation was observed between the serum FABP4 and triglycerides levels in the GDM group. The urine FABP4 concentration correlated positively with the BMIs before pregnancy and at delivery as well as with the urine ghrelin level in the GDM group. A negative correlation was noted between the urine FABP4 and BCMI in this group (Table 3).

The Benjamini-Hochberg procedure with a false discovery rate of 0.1 revealed that all of the originally significant associations were still significant.

## 4. Discussion

Adipokines are involved in many physiological processes and might contribute to pregnancy complications such as GDM and EGWG. They also seem to play a significant role in the pathogenesis of T2DM, obesity, and metabolic syndrome in the future maternal life. The adipose tissue secretes adipokines that can induce a pro-inflammatory environment. These cytokines are associated with the development of insulin resistance and metabolic abnormalities in local and peripheral tissues [28].

Our study showed that the highest serum FABP4 levels were present in the GDM mothers. The serum leptin concentrations were the lowest in the healthy women. These results are consistent with the observations made by other authors [11,29,30]. FABP4 and leptin are indicated as prime candidates for direct involvement in the pathophysiology of GDM and its long-term post-partum complications. Placental and non-placental origins of these adipokines are likely to contribute to dysregulated concentrations in GDM by induction of insulin resistance or β-cell dysfunction [11].

Of the more recently discovered pro-inflammatory adipokines, the insulin resistance-inducing FABP4 is an interesting candidate, which seems to be involved in the pathophysiology of GDM. FABP4 belongs to the fatty acid-binding protein superfamily and is highly expressed in adipocytes, macrophages and endothelial cells. This adipokine is known to impair glucose tolerance in mice by upregulating hepatic glucose production. FABPs, a family of intracellular lipid chaperones, are small (15 kDa) cytosolic, highly-affinic proteins that can reversibly bind to hydrophobic ligands, such as unesterified long-chain fatty acids, and coordinate lipid responses in the cells [23,31]. FABPs have been suggested to facilitate the transport of lipids to specific compartments in the cell [24].

It has been reported that increased circulating FABP4 levels are associated with obesity, insulin resistance, T2DM, hypertension, breast cancer independent of obesity, cardiac dysfunction, atherosclerosis, and metabolic syndrome [21,32,33,34,35,36]. Circulating FABP4 concentrations were significantly increased in obese subjects compared to their level in the lean controls, and serum FABP4 level positively correlated with the waist circumference, blood pressure, and insulin resistance [33]. High concentration of FABP4 at baseline was an independent predictor for the development of metabolic syndrome during a five-year follow-up period in a Chinese population [34]. A 10-year prospective study also showed that an increase of FABP4 level independently predicted the development of T2DM [35]. Circulating FABP4 is not only a potent biomarker but, as an adipokine, it also plays an important role in the development of metabolic syndrome and cardiovascular diseases [21]. FABP4 is thought to be a marker of pathophysiology and could be a treatment-target in T2DM [37].

Previous studies found that the serum FABP4 concentrations were significantly increased in the GDM pregnant women when compared with the controls [29,38,39]. Li et al. explained high circulating FABP4 levels in the maternal serum of pregnant GDM women by an increased FABP4 originating from the placenta and adipocytes [10,36]. Moreover, candidates for the placental hormones to induce FABP4 overexpression in the placenta and decidua in GDM include human placental lactogen and progesterone. Both increase continuously until term and may be associated with an increased insulin resistance together with the gestational age [36,40]. On the other hand, the serum from the pregnant GDM women affects adipocyte differentiation and proliferation, which will result in an increased level of FABP4. Expression of FABP4 mRNA in the placenta and decidua of the GDM patients is greater than that in the normal tissues [36]. The synergistic effects of FABP4 from the placenta and adipocytes can act on the metabolic and inflammatory pathways via adipocytes. These activities may play crucial roles in the development of insulin resistance and T2DM [36].

In the follow-up study of women six years after GDM, Svensson et al. [41] identified three factors —all related to the adipose tissue and body composition—that, independently of BMI and ethnicity, may increase the risk of progression to T2DM following GDM. These factors included increased serum FABP4 levels, weight gain after index pregnancy and a lower proportion of the fat-free mass. High BMI, abdominal fat distribution and enlarged adipocytes with corresponding signs of inflammation were also associated with the development of T2DM after GDM [41].

Our study is probably the first to report the urine FABP4 levels in the early puerperal mothers, yet we did not find any significant differences in this parameter between the healthy, EGWG, and GDM women. Urine, as a material for the assessment of FABP4 concentrations in the human studies, has been used in patients with glomerular injury, as well as with acute and chronic renal dysfunction [23,24,25]. Due to its small molecular size, this protein may escape the glomerular sieves and be excreted in the urine if not reabsorbed by the renal tubules. FABP4 may also be produced by the tubular epithelial cells or adjacent tissue [42]. In the healthy kidneys, FABP4 is expressed in the endothelial cells of the tubulointerstitial peritubular capillary and vein in both the cortex and medulla, but not in the glomerular or arterial endothelial cells, under normal physiological conditions [24,43]. FABP4 was remarkably expressed in the glomerulus in the patients with strong inflammatory capillary disorders [24,44]. Furthermore, the level of FABP4 expression in the glomerulus was significantly higher in the patients with endothelial proliferative lesions than in those without the lesions in IgA nephropathy. These findings indicate that ectopic expression of FABP4 in the glomerular endothelial cells can be associated with a local inflammation in the glomerulus [24]. Moreover, Obajdin et al. [25] hypothesized that increased urine FABP4 levels were not the result of enhanced de novo synthesis of FABP4 mRNA but rather of an abnormal loss of FABP4 by the cells of the loop of Henle. As a low molecular weight protein (15 kDa), FABP4 should be freely filtered in the glomerulus and then reabsorbed by the proximal tubules under normal conditions [45].

Our findings revealed that the serum and urine FABP4 concentrations correlated negatively in the healthy puerperal mothers. Okazaki et al. [24] noted that elevated levels of urine FABP4 were associated with glomerular damage and progression of proteinuria but only a weak positive correlation was seen between the urinary and serum FABP4. Unfortunately, although the study exclusion criteria comprised abnormalities in the GFR and urine test, particularly in proteinuria, the methodology of our study did not include more specialized techniques, such as electrophoresis and flow cytometry of urine proteins. It should also be emphasized that future evaluation of adipokines in the urine should be adjusted by the urine creatinine concentrations or daily excretion rate.

Moreover, the control group also provided significant evidence for the existence of a direct correlation between the urine FABP4 and leptin levels. Interestingly, positive correlations were found between the urine FABP4 and serum leptin levels in the EGWG group as well as between the serum FABP4 and leptin concentrations in the GDM subjects. Unfortunately, due to the obtained leptin levels in the urine in the EGWG and GDM mothers which were below the threshold of the sensitivity of the ELISA test, this parameter was not taken into consideration in these two groups for the correlating.

Ortega-Senovilla et al. [39] observed significant correlations between the FABP4 and leptin levels in the maternal blood in both control and GDM pregnant patients. In our study similar observations were made only in the GDM group since a direct correlation between these adipokines was revealed only in the urine of healthy mothers. It should be emphasized that in contrast to the cited study in which the maternal blood was collected before delivery, our study was performed in the early puerperium period [39].

In our study positive correlations were found between the serum FABP4 and urine ghrelin as well as between the urine FABP4 and serum ghrelin levels in the EGWG group and also between the urine FABP4 and ghrelin levels in the GDM group. Our previous study showed that women with GDM had lower serum and higher urine ghrelin concentrations in the early post-partum period in comparison with the healthy controls, i.e., at 48 h after delivery, yet, these differences were statistically insignificant [2]. On the other hand, there was a positive correlation between the serum and urine ghrelin levels, but only in the healthy mothers. This association between the ghrelin concentrations in these two biological materials was not observed in the GDM group. It seems that this irrelevant difference in favour of greater ghrelin levels in urine might be caused by disturbed metabolism of the circulating ghrelin in GDM mothers. This might have resulted from increased renal ghrelin clearance [2]. Zhang et al. demonstrated that overexpression of ghrelin gene via the FABP4 promoter exhibits high plasma concentration of des-acyl ghrelin, alters the development of the white adipose tissues, and improves glucose tolerance and insulin sensitivity [46].

In our study, the patients of GDM group were the oldest and were characterized by the highest values of BMIs evaluated before pregnancy. It is worth noting that GDM women are usually older and overweight or obese [15,16]. Several case-control studies have found that increased insulin resistance during pregnancy is associated with abnormalities in the body weight [10,47,48]. Higher pre-pregnant BMI is connected with an increased risk of GDM [49]. A positive relationship was noted between gestational age, pre-pregnant BMI, and HOMA-IR in the GDM patients [10].

The largest gestational weight and BMI gains were observed in the EGWG group, which was a consequence of the choice of inclusion criteria for this study group. Despite the fact that a higher BMI loss at 48 h after delivery was observed in this group in comparison with the GDM group, the BMI gain in the period from pre-pregnancy to 48 h after delivery (ΔBMI), i.e., the difference between the post-partum and pre-pregnancy BMIs, was the largest in the EGWG mothers. ΔBMI seems to be the most meaningful parameter of the body change for the future maternal life. Gestational weight gain in excess of the recommended IOM guidelines is now understood to be particularly important for setting the stage for subsequent health post-partum [50]. EGWG is currently considered as the first step in the vicious cycle and is arguably the most deleterious consequence of pregnancy impacting the future health of the mother. There are several reports including an extensive meta-analysis in over 69,000 women that indicate the excess weight gained in pregnancy is still retained some 20 years later [51]. Not surprisingly, the degree of the weight retained post-partum is highly variable between women, and often dependent on the amount of gestational weight gain, however, women with EGWG retain the most weight post-partum [50,51].

The evaluation of associations between FABP4 and the body weight measurements, BMIs and their changes in the period between pre-pregnancy to 48 h after delivery revealed that the serum FABP4 level directly correlated only with the pre-pregnant BMI in the healthy mothers, while their urine FABP4 was negatively associated with pre-BMI, BMI at delivery, and at the time of sampling. It is worth noticing the correlation at this last point of time, i.e., at 48 h after delivery, which turned out to be highly statistically significant.

The existence of a relationship between both serum and urine FABP4 and BMI confirms the correlation with the post-partum reduction of BMI (i.e., ΔBMI-2), as we presumed, which was negative with the serum FABP4 and positive with the urinary FABP4. It seems that the smaller decrease in BMI after delivery, the higher the FABP4 concentration in the serum and lower in the urine. The results of the correlations of the urine FABP4 with BMIs are especially noteworthy. However, the existence of the relationship between FABP4 levels and BMIs was not so obvious in the EGWG and GDM groups. Negative correlations were found only between the FABP4 serum levels and gestational weight and BMI gains, post-partum BMI, and BMI change after pregnancy and delivery (i.e., ΔBMI) in the EGWG group. In the GDM group, the FABP4 level positively correlated in the serum with the BMIs at delivery and after delivery as well as in the urine with the BMIs before pregnancy and at delivery. The results in these two groups are probably related to the fact that the women in GDM group, due to pre-pregnant overweight and obesity and with normal carbohydrate results in the first trimester, remained under strict control, so that from 24–28 weeks of pregnancy after being diagnosed with GDM they were successfully treated with not only a proper diet but also with insulin. Meanwhile, due to the normal body mass and BMIs before pregnancy and with normal OGTT results at 24–28 weeks of pregnancy, the patients in the EGWG group were not under accurate dietary control throughout their pregnancy.

Ortega-Senovilla et al. [39] was able to find significant correlations between FABP4 in the maternal blood and pre-pregnancy BMI in both control and GDM pregnant patients [39]. However, our study was performed two days after delivery, i.e., two days after the placenta delivery, whereas Ortega-Senovilla et al. evaluated the serum FABP4 levels in each mother at the last visit to the obstetric clinic, no longer than one week before delivery [39]. Taking into account that FABP4 concentrations during pregnancy are partially related to the placental factors, our results cannot be directly compared with the cited study.

This study has demonstrated that healthy mothers had higher levels of total cholesterol, HDL, and LDL than the GDM women. These results are consistent with the observations made by other authors [10,52]. It has been documented that women with a history of GDM exhibit altered CVD risk factors, including lower HDL concentrations, when compared with mothers with healthy pregnancies [7,8]. It is worth noting that HDL represents approximately 20% of the total plasma cholesterol and is inversely related to the occurrence of CVD. In the women without GDM, the HDL levels decrease during the initial 12 weeks post-partum, but the magnitude of this decline is small. Moreover, it has been shown that during pregnancy, women with GDM have lower levels of HDL than those without GDM [53].

Dubé et al. [52] found decreased maternal plasma levels of total cholesterol, LDL, and HDL, which is in agreement with ours and other authors’ studies [42,43], and they concluded that these results could partly be due to the lack of excessive gestational weight gain in obese women and/or to the unchanged levels of triglycerides and free fatty acids in the venous cord blood despite the increased maternal plasma levels of triglycerides in obese pregnancies. Studies have reported mostly inflammatory changes in the placenta of obese women, such as an accumulation of the immune cells (macrophages and neutrophils) and an increase in the expression of several inflammatory cytokines, such as interleukin-1 (IL-1), IL-6, TNFα, IL-8 [52,54,55].

Our findings revealed that the serum FABP4 concentrations correlated positively with the lipid profile (i.e., total cholesterol, HDL, and LDL levels), as well as negatively with HgbA1c concentrations in the healthy puerperal mothers. Negative correlations were found between the urine FABP4 and albumin and LDL levels in this group. It should be noted that, in our study, all abnormalities in the urine analysis, including proteinuria, were excluded.

No significant correlations were found in either urine or serum FABP4 levels with biochemical findings in the EGWG group. A negative correlation was observed between the serum FABP4 and triglycerides levels only in the GDM group.

As an important intracellular fatty acids carrier protein, FABP4, is released from adipocytes and plays an important biologic role in the fatty acid uptake, transport, and metabolism [10]. FABP4 may influence insulin sensitivity and energy metabolism by regulation of fatty acid metabolism. Increased FABP4 levels may promote the accumulation of short-chain fatty acids in the cells and decrease PI3K-AKT protein activity, thereby inhibiting glucose oxidation and glycolysis and significantly reducing glucose uptake and utilization in the muscle and liver [10,35,56]. Therefore, the pathway from glucose to triglycerides is disturbed and the increased insulin resistance may lead to GDM [10,35,56]. On the other hand, it should also be emphasized that future studies should take into account the impact of insulin treatment in patients with GDM on the serum and urine levels of FABP4. In our study, since in all the included patients with GDM the same modality of treatment with insulin was used, we did not decide for an assessment of insulin impact on the serum and urine levels of FABP4. Furthermore, what is also extremely important is the time interval between the last insulin dose and the FABP4 level measurement. In our study, all of the patients with a GDM history received the last insulin injection at least 48 h earlier.

In order to assess the maternal body composition and hydration status, the BIA method was used in the study. This standardized technique is non-invasive, fast, and well tolerated by patients [57,58,59]. The physical properties of BIA, its measurement variables, and their clinical significance, have well been described in many previously published reports [58,60]. BIA appears to be a more accurate predictor of the gestational and post-partum outcomes than BMI [61]. Wang et al. [61] noted that the maternal increased BMI and gestational weight gain reflect the pregnancy nutritional status. BMI, however, is only a surrogate indicator of obesity and does not measure the distribution of fat. Fat and free-fat masses, measured by BIA, can accurately reflect the body fat compositions and have been considered as better predictors of maternal nutritional status than BMI [61].

Very few studies concerning BIA in the assessment of GDM have been reported so far. Our previous study showed that mothers with GDM in the early puerperium, when compared with the healthy controls, presented higher levels of not only FTI, which is defined as the adipose tissue mass divided by the square of the body height and is expressed in units of kg/m^2^, but also of TBW and ECW, where the latter consists of the interstitial water, plasma water, and transcellular water [2].

In the current study, the lowest results of analysed BIA parameters (i.e., TBW, ECW, and FTI) were observed in the healthy subjects. The EGWG group was characterized by the highest values regarding ICW between all the studied groups as well as higher values of LTI and BCMI in comparison to the healthy mothers. The aforementioned results suggest that the EGWG women presented a higher degree of disturbances in the hydration status and body composition in the early puerperium.

Gilmore et al. [50] emphasized that though total post-partum weight retention is the most frequently reported variable and fewer data on the composition of retained weight are available, it should be noted that the distribution of the post-partum weight being free-fat mass or fat mass is of particular important. Body composition changes after delivery may be a tell-tale sign of future metabolic health problems. During the early post-partum period the demand for expanded extracellular and intracellular fluid, mammary tissue (if not breastfeeding) and uterine tissue is no longer present, so the weight attributed to these areas naturally decreases and will return to the preconception status usually within 6–12 weeks [50]. Despite the weight loss seen from those tissues, the total fat mass is increased post-partum by 4% and visceral fat by 33% above the preconception values [62]. Women with EGWG have three times greater incidence of abdominal obesity at eight years post-partum compared with those who gained the recommended gestational weight [63]. Some authors observed the perpetuating cycle of weight gain in pregnancy and post-partum and amplification of both outcomes with subsequent pregnancies and pointed to the detrimental impact of pregnancy and excess GWG on the visceral fat accumulation. Just as the accumulation of visceral fat is understood to increase the risk of pregnancy-related adverse outcomes, such as GDM, the increased abdominal adiposity post-partum contributes to the progression of impaired glucose tolerance in pregnancy, to insulin resistance and, finally, to T2DM post-partum as well as other co-morbidities, such as CVD and metabolic syndrome, as the woman is aging [50,64].

Evaluation of associations between the FABP4 and BIA variables revealed negative correlations between the urine FABP4 levels and almost all BIA findings with the exception of ECW which consists of the interstitial water, plasma water and transcellular water only in the healthy subjects. Meanwhile, the FABP4 concentrations were associated negatively with ECW in the serum, as well as with LTI and BCMI in the urine of EGWG patients. The serum FABP4 levels correlated positively with TBW, ICW, and FTI, whereas the urine FABP4 concentrations were negatively related to BCMI in the GDM group.

Interestingly, there was only one BIA parameter, i.e., BCMI, which was negatively correlated with the urine FABP4 concentrations in all the studied groups. This parameter was introduced by us taking into account the difference in the BMI in the post-partum period between our patients and it was calculated as the body cell mass divided by the square of the body height as it seems to be a more precise parameter than BCM.

Gilmore et al. [50] concluded that pregnancy and the post-partum periods are not independent, but highly interrelated and affect one another. Many of the physiological and anatomical changes during gestation, particularly those to support the expansion of the fat mass, appear to prepare the mother for the energy-costly process of lactation. However, this turns out now to be a double-edged sword as the weight gain and metabolic changes that occur during pregnancy, if not controlled, can also have negative consequences for the post-partum woman and increase the risk of developing chronic disease later in her life [50].

## 5. Conclusions

In summary, we were able to show that the serum FABP4 levels were significantly the highest in the GDM group in the early puerperium. On the basis of our findings and previous studies it appears that increased circulating FABP4 concentrations can persist in GDM patients after delivery and might contribute to the increased risk of T2DM and metabolic syndrome. On the other hand, evaluation of FABP4 may be used as a predictive marker for mothers with the history of GDM.

The study results obtained from the observed correlations of FABP4 (in the serum and urine) and biophysical and biochemical variables confirm that further studies should not be limited to evaluating only the serum FABP4 levels. It seems that urine should also be taken into account since this material may provide additional information and help to explain the pathogenesis of T2DM.

It also seems understandable that obstetricians and perinatologists should pay attention to nutrition and weight management during pregnancy, educate pregnant women that their overnutrition predisposes them to metabolic disorders later in life. Our study may confirm that EGWG, seen as an obstetric complication, is associated with disturbances in the body composition and hydration status.

Nonetheless, the physiological and pathological significance of these findings requires further elucidation.

## Figures and Tables

**Figure 1 jcm-07-00505-f001:**
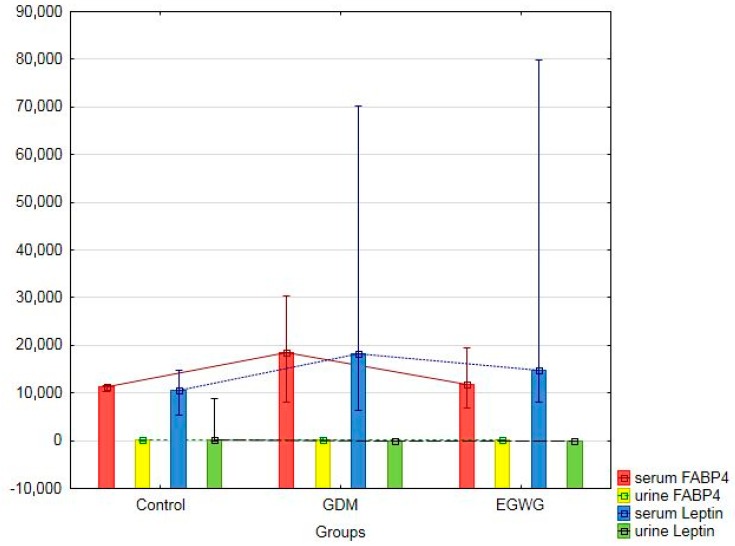
FABP4 and leptin levels in the serum and urine in the control, GDM, and EGWG groups.

**Figure 2 jcm-07-00505-f002:**
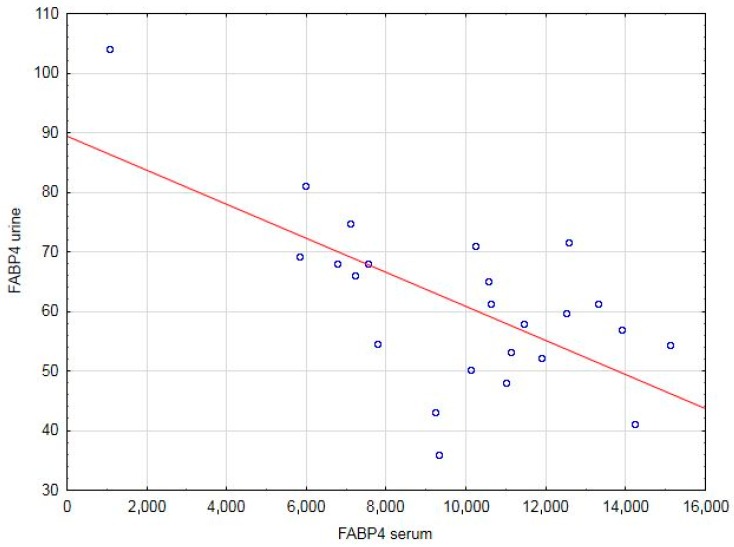
A scatterplot diagram showing a negative correlation between the serum and urinary FABP4 concentrations in the healthy group.

**Table 1 jcm-07-00505-t001:** Comparison of characteristics of the subjects.

Variables	Control Group (*n* = 24)	EGWG Group (*n* = 24)	GDM Group (*n* = 22)	*p*
**Pregnancy & Delivery**				
Age (years)	30.5 (30–36)	33.5 (28–35)	36.1 (32–41)	**0.01 ***
Pre-pregnancy BMI (kg/m^2^)	21.77 (20.2–24.4)	23 (22.3–24.45)	28.1 (26.6–30.1)	**0.00001 *****
Gestational weight gain (kg)	14.5 (8–15)	22.75 (21.5–25)	11 (7–15)	**0.00001 *****
ΔBMI-1 (kg/m^2^)	5.44 (2.97–5.6)	7.75 (7.63–8.8)	3.9 (2.23–5.19)	**0.00001 *****
BMI at delivery (kg/m^2^)	26.4 (25.1–29.1)	30.73 (29.58–32)	32.2 (30.5–33.9)	**0.00001 *****
**2nd Day of Post-Partum Period**				
Post-partum BMI (kg/m^2^)	22.9 (21–23.9)	27.95 (26.6–29.2)	29.8 (27.7–31.2)	**0.00001 *****
ΔBMI-2 (kg/m^2^)	2.53 (2.08–4.16)	2.79 (2.35–3.20)	2.2 (1.28–2.7)	**0.0129 ***
ΔBMI (kg/m^2^)	0.51 (–1.23–2,6)	5.3 (4.92–5.7)	2,2 (0.1–2,9)	**0.00001 *****
Albumin (g/dL)	3.68 (3.43–3.73)	3.57 (3.35-3.77)	3.46 (3.37–3.64)	0.0689
Total cholesterol (mg/dL)	257 (207–287)	225.5 (196–246)	199.5 (192–242)	**0.0127 ***
HDL (mg/dL)	78 (75–79)	68.5 (58.5–77.5)	65 (54–73)	**0.0066 ***
LDL (mg/dL)	131 (102–152)	104.5 (85–125.5)	100 (84–127)	**0.0057 ***
Triglycerides (mg/dL)	190 (150–254)	198.5 (183–253.5)	240.5 (155–252)	0.3797
Haemoglobin A1c (%)	5.35 (5.2–5.4)	5.5 (5.15–5.5)	5.6 (5.2–5.8)	0.0814
Serum ghrelin (ng/mL)	0.93 (0.65–1.11)	1.19 (0.34–2.43)	0.4 (0.19–1.23)	0.0624
Urine ghrelin (ng/mL)	0.1 (0.1–0.29)	0.12 (0.04–0.30)	0.21 (0.07–6.6)	0.3292
Serum FABP4 (pg/mL)	11,179 (1062–11,563)	11,725 (10,203–15,480)	18,463 (13,321–24,291)	**0.0006 ****
Urine FABP4 (pg/mL)	61 (31–103.2)	39.7 (18.6–77.7)	66.7 (22.9–338.9)	0.192
Serum leptin (pg/mL)	10,429 (6044–14,884)	14,873 (12,591–47,582)	18,154 (10,836–51,914)	**0.0011 ***
Urine leptin (pg/mL)	162.3 (0–4013.1)	ND	ND	**-**
Total body water (L)	29.9 (26–32.9)	35.3 (33.9–40.6)	33.9 (32.3–36.2)	**0.00001 *****
Extracellular water (L)	14.5 (13–15.7)	16.9 (15.7–20)	16.6 (15–17.8)	**0.00001 *****
Intracellular water (L)	15.4 (13.5–17.1)	19.05 (17.8–20)	17.3 (16.8–18.4)	**0.00001 *****
Lean tissue index (kg/m^2^)	10 (9.4–13.1)	13 (12.4–14.1)	12.1 (11–13.2)	**0.0013 ***
Fat tissue index (kg/m^2^)	11.1 (9.9–13.8)	14 (13.3–14.9)	17 (13.3–18.8)	**0.00001 *****
BCMI (kg/m^2^)	5.1 (4.8–7.2)	6.97 (6.5–7.9)	6.2 (5.5–7.3)	**0.0008 ****

The results are shown as the median (interquartile range 25–75%). Statistically significant values are given in the bold type. * *p* < 0.05; ** *p* < 0.001; *** *p* < 0.0001. BCMI—body cell mass index; BMI—body mass index; ΔBMI-1—gestational BMI gain; ΔBMI-2—BMI loss at 48 h after delivery; ΔBMI—BMI gain in the period from pre-pregnancy to 48 h after delivery; EGWG—excessive gestational weight gain; FABP4—fatty acid binding protein 4; GDM—gestational diabetes mellitus; HDL—high-density lipoprotein cholesterol; LDL—low-density lipoprotein cholesterol; ND—detected below the threshold of the sensitivity of the ELISA test.

**Table 2 jcm-07-00505-t002:** Comparison of characteristics of the subjects—the post-hoc analysis.

Variables	KW	Control & EGWG	Control & GDM	EGWG & GDM
Age	**0.01 ***	1.0	**0.018 ***	**0.036 ***
Pre-pregnancy BMI	**0.00001 *****	0.694	**0.000003 *****	**0.0006 ****
Gestational weight gain	**0.00001 *****	**0.0000001 *****	0.889	**0.0000001 *****
ΔBMI-1	**0.00001 *****	**0.0000001 *****	0.605	**0.0000001 *****
BMI at delivery	**0.00001 *****	**0.00017 ****	**0.000001 *****	0.687
Post-partum BMI	**0.00001 *****	**0.00005 *****	**0.000002 *****	1.0
ΔBMI-2	**0.0129 ***	1.0	0.112	**0.013 ***
ΔBMI	**0.00001 *****	**0.000002 *****	0.467	**0.003 ***
Total cholesterol	**0.0127 ***	0.151	**0.012 ***	0.941
HDL	**0.0066 ***	0.136	**0.006 ***	0.691
LDL	**0.0057 ***	**0.028 ***	**0.011 ***	1.0
Serum FABP4	**0.0006 ****	1.0	**0.002 ***	**0.003 ***
Serum leptin	**0.0011 ***	**0.007 ***	**0.003 ***	1.0
Total body water	**0.00001 *****	**0.000004 *****	**0.004 ***	0.397
Extracellular water	**0.00001 *****	**0.00002 *****	**0.0002 ****	1.0
Intracellular water	**0.00001 *****	**0.000002 *****	0.095	**0.022 ***
Lean tissue index	**0.0013 ***	**0.0008 ****	0.187	0.274
Fat tissue index	**0.00001 *****	**0.003 ***	**0.00002 *****	0.576
BCMI	**0.0008 ****	**0.0005 ****	0.215	0.176

Statistically significant values are given in the bold type. * *p* < 0.05; ** *p* < 0.001; *** *p* < 0.0001. KW—Kruskal-Wallis analysis of variance; BCMI—body cell mass index; BMI—body mass index; ΔBMI-1—gestational BMI gain; ΔBMI-2—BMI loss at 48 h after delivery; ΔBMI—BMI gain in the period from pre-pregnancy to 48 h after delivery; EGWG—excessive gestational weight gain; FABP4—fatty acid binding protein 4; GDM—gestational diabetes mellitus; HDL—high-density lipoprotein cholesterol; LDL—low-density lipoprotein cholesterol.

**Table 3 jcm-07-00505-t003:** Correlations between the maternal serum FABP4 levels and urine FABP4 levels and the selected parameters.

Variables	Healthy Group		EGWG Group		GDM Group	
	Serum FABP4	Urine FABP4	Serum FABP4	Urine FABP4	Serum FABP4	Urine FABP4
Pre-pregnancy BMI	**0.60 ***	**−0.60 ***	0.203	0.101	0.382	**0.527 ***
Gestational weight gain	−0.174	−0.290	**−0.750 *****	0.014	0.119	0.123
ΔBMI-1	0.028	−0.257	**−0.487 ***	−0.080	−0.409	−0.082
BMI at delivery	0.371	**−0.486 ***	−0.207	0.371	**0.600 ***	**0.733 ****
Post-partum BMI	0.20	**−0.771 *****	**−0.539 ***	0.154	**0.576 ***	0.370
ΔBMI-2	**−0.543 ***	**0.543 ***	0.355	0.239	−0.028	0.065
ΔBMI	−0.143	**−0.543 ***	**−0.802 *****	0.112	0.176	−0.103
Albumin	0.371	**−0.943 *****	0.119	−0.112	−0.291	−0.145
Total cholesterol	**0.899 *****	−0.377	0.077	−0.168	−0.224	0.261
HDL	**0.667 ****	−0.377	0.014	−0.175	−0.054	−0.079
LDL	**0.943 *****	**−0.486 ***	0.015	−0.112	−0.151	−0.164
Triglycerides	0.348	−0.116	0.301	−0.007	**−0.564 ***	0.382
Haemoglobin A1c	**−0.406 ***	−0.116	0.388	−0.011	−0.221	−0.129
Serum ghrelin	−0.314	0.314	−0.280	**0.517 ***	−0.027	−0.036
Urine ghrelin	0.028	0.314	**0.447 ***	0.336	−0.045	**0.518 ***
Urine FABP4	**−0.543 ***	-	0.168	-	0.209	-
Serum leptin	−0.314	−0.143	−0.182	**0.447 ***	**0.845 *****	0.418
Urine leptin	−0.334	**0.941 *****	-	-	-	-
Total body water	−0.086	**−0.486 ***	−0.294	−0.126	**0.427 ***	−0.118
Extracellular water	−0.314	−0.371	**−0.444 ***	0.129	0.373	−0.118
Intracellular water	0.086	**−0.771 *****	−0.049	−0.169	**0.484***	−0.265
Lean tissue index	−0.028	**−0.543 ***	0.059	**−0.501 ***	−0.383	−0.332
Fat tissue index	0.314	**−0.543 ***	−0.301	0.133	**0.573 ***	0.282
BCMI	−0.086	**−0.486 ***	0.015	**−0.410 ***	−0.391	**−0.482 ***

Statistically significant values are given in the bold type. * *p* < 0.05; ** *p* < 0.001; *** *p* < 0.0001. BCMI—body cell mass index; BMI—body mass index; ΔBMI-1—gestational BMI gain; ΔBMI-2—BMI loss at 48 h after delivery; ΔBMI—BMI gain in the period from pre-pregnancy to 48 h after delivery; EGWG—excessive gestational weight gain; FABP4—fatty acid binding protein 4; GDM—gestational diabetes mellitus; HDL—high-density lipoprotein cholesterol; LDL—low-density lipoprotein cholesterol.

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
