# Peer review of "Fatty Acid-Binding Protein 4—An “Inauspicious” Adipokine—In Serum and Urine of Post-Partum Women with Excessive Gestational Weight Gain and Gestational Diabetes Mellitus"

_jcm, 2018, doi:10.3390/jcm7120505_

Round 1
Reviewer 1 Report
Zaneta Kimber-Trojnar et al investigated the serum and urine concentration of FABP4 concentrations in normal, GDM and EGWG women. Healthy women were characterized by lowest leptin levels and a negative correlation between serum and urine FABP4 levels. Serum FABP4 levels were the highest in GDM group and serum FABP4 and leptin correlated positively. EGWG group demonstrated the positive correlations between urine FABP4 and serum leptin and ghrelin concentrations.
Although the adipokine concentrations were extensively measured, it is rather descriptive and lacks in the insights into the pathomechanism of GDM.
Major comments
1. In lines 100-126, the authors should indicate how many pregnancy women were recruited, how many women were excluded from the study. The number of three categories, normal glucose tolerance, EGWG, and GDM, are similar by chance?
2. In lines 127-136, urine concentration of adipokines should be adjusted by urine creatinine concentrations (gCr) or daily excretion rate should be examined.
3. In Figure 1, urinary concentrations of lectin and FABP4 should be shown.
4. In Table 3, why serum and urinary FABP4 were negatively correlated? In addition, scatter diagram should be shown.
Author Response
Dear Sir,
We would like to express our gratitude for your meaningful and helpful comments which have made a substantial contribution to the quality of our paper.
Having studied the Reviewers’ comments and following your advice, we have decided to introduce some changes into our paper, which, we do hope, will bring some improvement to our manuscript.
Our responses to the Reviewer's comments:
“Although the adipokine concentrations were extensively measured, it is rather descriptive and lacks in the insights into the pathomechanism of GDM.”
The aim of the study was to focus on women in the early post-partum period with a history of two complications associated with their recent pregnancies, i.e. gestational diabetes mellitus or excessive gestational weight gain. According to the current state of scientific knowledge, these problems have been regarded as independent risk factors of chronic diseases later in their lives. It has been observed that these women are much more prone to suffer from type 2 diabetes mellitus, obesity, cardiovascular diseases, and metabolic syndrome in the future. We chose a new adipokine: fatty acid binding protein 4 (FABP4), which seems to be “inauspicious” to compare with results of well-known pro-inflammatory leptin.
On the basis of our findings it appears that FABP4 concentrations are highly elevated in GDM females after delivery. Increased expression of FABP4 might contribute to the increased risk of T2DM and metabolic syndrome, as it has been suggested in previous studies. Thus, evaluation of FABP4 may be used as a predictive marker of future diseases for these women.
“Major comments
1. In lines 100-126, the authors should indicate how many pregnancy women were recruited, how many women were excluded from the study. The number of three categories, normal glucose tolerance, EGWG, and GDM, are similar by chance?”
During long data collection (between March 2016 and February 2017) there were 2.519 deliveries in our Department. For reducing the number of factors that could distort results, we had to exclude all preterm deliveries (before 37 weeks of pregnancy), multiple pregnancies and all patients with any complications/diseases except gestational diabetes mellitus or excessive gestational weight gain (at least 20 kg during pregnancy). We had to rule out among others all patients with gestational hypertension, which is a common complication observed in women with diabetes or obesity.
We have modified the sentence:
“All of the study subjects included in this study were Caucasian and they were divided into three groups. We selected three groups of patients.”
We have provided additional details concerning the inclusion criteria:
“The second group included 24 patients with EGWG, i.e. with normal pre-pregnancy BMI (i.e. between 18.5 and 24.99 kg/m2), three normal results of the OGTT at 24-28 weeks of gestation and gestational weight gain of at least 20 kilograms. This subgroup had no concomitant diseases and received only vitamin-iron supplementation.
The third studied group consisted of 22 patients with diagnosed GDM who were on a diet and receiving insulin treatment. 27% of the GDM subjects were treated with intensive insulin therapy, while 73% of them were controlled with only one basal insulin injection per day. Diagnostic criteria for GDM were based on the OGTT at 24-28 weeks of gestation: fasting glucose ≥ 5,1mmol/l (92 mg/dl), or one hour plasma glucose result of ≥ 10.0 mmol/l (180 mg/dl), or a two-hour plasma glucose result of ≥ 8,5 mmol/l (153 mg/dl) [24,25]. This subgroup had no concomitant diseases and received only vitamin-iron supplementation and anti-diabetic treatment.”
2. “In lines 127-136, urine concentration of adipokines should be adjusted by urine creatinine concentrations (gCr) or daily excretion rate should be examined.”
We fully agree with the Reviewer that we did not examine urine creatinine concentrations / daily excretion rate. However, we excluded from the study females with current urinary infections, abnormal laboratory results (e. g. urine test parameters, creatinine serum level and glomerular filtration rate, BUN, uric acid), chronic renal diseases, arterial hypertension, abnormal serum electrolytes (including hypokalemia and hypercalcemia). Also, we did not observe any statistical differences in above mentioned measurable parameters (within their normal ranges) between GDM patients, EGWG patients, and controls (data not shown).
On the other hand, the Reviewer's suggestion that urine concentration of adipokines should be adjusted by urine creatinine concentrations (gCr) or daily excretion rate is meaningful and helpful. We will take into account this comment during our future grant.
We have added the comment in the Discussion:
“It should also be emphasized that future evaluation of adipokines in the urine should be adjusted by the urine creatinine concentrations (gCr) or daily excretion rate.”
3. “In Figure 1, urinary concentrations of lectin and FABP4 should be shown.”
We have modified the Figure 1 with due regard to large differences in the concentrations of both leptin and FABP4 in the serum and urine. The minimum detectable dose of human leptin is typically less than 7.8 pg/mL. The urine leptin levels obtained in the majority of the EGWG and GDM patients were below the threshold of the sensitivity of the ELISA test.
4. “In Table 3, why serum and urinary FABP4 were negatively correlated? In addition, scatter diagram should be shown.”
We tried to explain a negative correlation between the serum and urinary FABP4 concentrations revealed in the healthy subjects in the Discussion. We realize that we found this correlation only in one group of 24 patients. We have added Figure 2 with a scatter plot diagram of this association.
We appreciate your time and look forward to your response.
Yours faithfully,
Zaneta Kimber-Trojnar and co-authors

Reviewer 2 Report
COMMENTS FOR AUTHORS: JMB FATTY ACID BINDING PROTEIN
Abstract and methods: Why was adiponectin not measured. Could it be measured in reserved serum?
Line 62: It would be valuable to measure adiponectin, since it is a prominent risk factor for GDM and is closely related to the measurements already made.
Line 97: delete the phrase “some disturbances” which is vague and non specific. Please specify which disturbances have led to this study, naming specific disturbances.
Was the study prospective? Were there any other inclusion criteria beyond at least 37 weeks gestation.
Were any other glucose regulating medications b VJT7GSNZPXDSMFD5esides insulin used in the GDM patients?
Given the multiple comparisons in Tables 1 and 2, the Benjamini-Hochberg correction for false positive results should be used.
Are serum and urine levels of FABP4 affected by insulin treatment for GDM? This may be a very important factor in the authors’ analyses
Author Response
Dear Sir,
We would like to express our gratitude for your meaningful and helpful comments which have made a substantial contribution to the quality of our paper.
Having studied the Reviewers’ comments and following your advice, we have decided to introduce some changes into our paper, which, we do hope, will bring some improvement to our manuscript.
Our responses to the Reviewer's comments:
“Abstract and methods: Why was adiponectin not measured. Could it be measured in reserved serum?
Line 62: It would be valuable to measure adiponectin, since it is a prominent risk factor for GDM and is closely related to the measurements already made.”
We have modified the sentence in our “Introduction”:
“In this study, we concentrated on the fatty acid binding protein 4 (FABP4). This a newer adipokine , which appears to be one of the most probable candidates involved in the pathophysiology of GDM besides , similarly to well-known and verified markers of diabetes and obesity such as adiponectin, leptin, and tumour necrosis factor α (TNFα) [11-13].”
We have also added two references to this sentence and changed the order of references:
“12. López-Tinoco, C.; Roca, M.; Fernández-Deudero, A.; García-Valero, A.; Bugatto, F.; Aguilar-Diosdado, M.; Bartha, J.L. Cytokine profile, metabolic syndrome and cardiovascular disease risk in women with late-onset gestational diabetes mellitus. Cytokine 2012, 58, 14-19. doi: 10.1016/j.cyto.2011.12.004.
13. Świrska, J.; Zwolak, A.; Dudzińska M.; Matyjaszek-Matuszek, B.; Paszkowski, T. Gestational diabetes mellitus - literature review on selected cytokines and hormones of confirmed or possible role in its pathogenesis. Ginekol. Pol. 2018, 89, 522-527. doi: 10.5603/GP.a2018.0089.”
We chose a new adipokine: fatty acid binding protein 4 (FABP4), which seems to be “inauspicious” to compare with results of well-known pro-inflammatory leptin. On the basis of our findings it appears that FABP4 concentrations are highly elevated in GDM females after delivery. Increased expression of FABP4 might contribute to the increased risk of T2DM and metabolic syndrome, as it has been suggested in previous studies. Thus, evaluation of FABP4 may be used as a predictive marker of future diseases for these women.
On the other hand, the Reviewer's conception about adiponectin, which might be measured in the reserved serum is meaningful and helpful. We will take into account this suggestion during our future grant.
“Line 97: delete the phrase “some disturbances” which is vague and non specific. Please specify which disturbances have led to this study, naming specific disturbances.”
The study aim has been modified. The phrase “some disturbances” has been removed. The names of specific disturbances have been added:
“The relationship between FABP4 and various biochemical and biophysical measurements in puerperal GDM and EGWG women is still awaiting explanation. We hypothesized that because of some disturbances, including both hyperglycemia first detected at 24-28 weeks of pregnancy and excess weight gain during pregnancy which are connecting with changes in the body composition and hydration status, the FABP4 concentrations in the serum and urine of women with GDM and EGWG in the early post-partum period would probably be impaired.”
“Was the study prospective? Were there any other inclusion criteria beyond at least 37 weeks gestation?”
“Was the study prospective? Were there any other inclusion criteria beyond at least 37 weeks gestation?”
Our study was not prospective. During long data collection (between March 2016 and February 2017) there were 2.519 deliveries in our Department. For reducing the number of factors that could distort results, we had to exclude all preterm deliveries (before 37 weeks of pregnancy), multiple pregnancies and all patients with any complications/diseases except gestational diabetes mellitus or excessive gestational weight gain (at least 20 kg during pregnancy). We had to rule out among others all patients with gestational hypertension, which is a common complication observed in women with diabetes or obesity.
We have modified the sentence:
“All of the study subjects included in this study were Caucasian and they were divided into three groups. We selected three groups of patients.”
We have provided additional details concerning the inclusion criteria:
“The second group included 24 patients with EGWG, i.e. with normal pre-pregnancy BMI (i.e. between 18.5 and 24.99 kg/m2), three normal results of the OGTT at 24-28 weeks of gestation and gestational weight gain of at least 20 kilograms. This subgroup had no concomitant diseases and received only vitamin-iron supplementation.
The third studied group consisted of 22 patients with diagnosed GDM who were on a diet and receiving insulin treatment. 27% of the GDM subjects were treated with intensive insulin therapy, while 73% of them were controlled with only one basal insulin injection per day. Diagnostic criteria for GDM were based on the OGTT at 24-28 weeks of gestation: fasting glucose ≥ 5,1mmol/l (92 mg/dl), or one hour plasma glucose result of ≥ 10.0 mmol/l (180 mg/dl), or a two-hour plasma glucose result of ≥ 8,5 mmol/l (153 mg/dl) [24,25]. This subgroup had no concomitant diseases and received only vitamin-iron supplementation and anti-diabetic treatment.”
“Were any other glucose regulating medications besides insulin used in the GDM patients?”
The GDM patients were on a diet and receiving insulin treatment. 27% of the GDM subjects were treated with intensive insulin therapy, while 73% of them were controlled with only one basal insulin injection per day. Our treatment of the GDM patients were based on the Guidelines of Polish Diabetes Association that gave Polish doctors strict General principles of diabetes treatment during pregnancy. These guidelines have included for a few years (cited):
"In many women with gestational diabetes, behavioral modifications allow adequate blood glucose control, and drug treatment using insulin should be initiated if therapeutic targets are not met." as well as
"Insulin is the only antidiabetic drug recommended in pregnancy. Based on the current knowledge, use of other antidiabetic drugs, either oral or GLP-1 receptor agonists, is not recommended." as well as
"Oral antidiabetic agents are currently not recommended for the treatment of diabetes during pregnancy due to the fact that they pass through the placenta and no data are available regarding their long-term effects in the offspring."
“Given the multiple comparisons in Tables 1 and 2, the Benjamini-Hochberg correction for false positive results should be used.”
According to the suggestion of the Reviewer, we have carefully revised this part of the Results section. The Benjamini-Hochberg procedure with a false discovery rate of 0.1 revealed that all of the originally significant associations are still significant.
We have refined the “Experimental Section”:
“The differences between the three studied groups were tested for significance using the Kruskal-Wallis analysis of variance. The post-hoc analysis of differences between two groups were tested for significance. The Spearman’s coefficient test was used for the correlation analyses. The Benjamini-Hochberg correction for false positive results was used.”
as well as “Results”:
“The Benjamini-Hochberg procedure with a false discovery rate of 0.1 revealed that all of the originally significant associations were still significant.”
“Are serum and urine levels of FABP4 affected by insulin treatment for GDM? This may be a very important factor in the authors’ analyses.”
Unfortunately, we cannot exclude that the serum and urine levels of FABP4 were affected by insulin treatment for the GDM patients. We agree wholeheartedly with our Reviewer that insulin might be a very important factor in our analyses.
Since in all the included patients with GDM the same modality of treatment with insulin was used, we did not decide for an assessment of insulin impact on the serum and urine levels of FABP4. Furthermore, what is also extremely important is the time interval between the last insulin dose and the FABP4 level measurement. In our study, all of the patients with a GDM history received the last insulin injection at least 48 h earlier.
This meaningful insight requires further elucidation by comparison of mothers with history of GDM who were controlled only by antidiabetic diet during pregnancy.
We have added the comment in the Discussion:
“On the other hand, it should also be emphasized that future studies should take into account the impact of insulin treatment in patients with GDM on the serum and urine levels of FABP4. In our study, since in all the included patients with GDM the same modality of treatment with insulin was used, we did not decide for an assessment of insulin impact on the serum and urine levels of FABP4. Furthermore, what is also extremely important is the time interval between the last insulin dose and the FABP4 level measurement. In our study, all of the patients with a GDM history received the last insulin injection at least 48 h earlier.”
We appreciate your time and look forward to your response.
Yours faithfully,
Zaneta Kimber-Trojnar and co-authors

Round 2
Reviewer 1 Report
No further comments